# Living with Infection Risk and Job Insecurity during COVID-19: The Relationship of Organizational Support, Organizational Commitment, and Turnover Intention

**DOI:** 10.3390/ijerph19148516

**Published:** 2022-07-12

**Authors:** Yusuf Yılmaz, Engin Üngüren, Ömer Akgün Tekin, Yaşar Yiğit Kaçmaz

**Affiliations:** 1Department of Recreation Management, Faculty of Tourism, Akdeniz University, Antalya 07070, Turkey; 2Department of Business Management, Faculty of Economics, Administrative and Social Sciences, Alanya Alaaddin Keykubat University, Antalya 07450, Turkey; engin.unguren@alanya.edu.tr; 3Department of Gastronomy and Culinary Arts, Manavgat Faculty of Tourism, Akdeniz University, Antalya 07600, Turkey; omeratekin@akdeniz.edu.tr; 4Alanya Alaaddin Keykubat University, Antalya 07450, Turkey; yykacmaz@hotmail.com

**Keywords:** COVID-19, organizational support, organizational commitment, turnover intention, job insecurity

## Abstract

The COVID-19 outbreak caused a stressful process for hospitality employees in terms of both being infected and experiencing the risk of losing their jobs. Stressful working conditions increase employees’ turnover intentions (TI). This study aims to analyze the relationship among perceived organizational support (POS), organizational commitment (OC), and turnover intention (TI) within the context of employees’ infection status and perceived job insecurity (JI). In this context, the study tests a moderated mediation research model. Having adopted a quantitative research method, data were acquired from 490 respondents who work at five-star accommodation companies in Alanya, Turkey. Findings show that the impact of POS on OC and IT differ according to employees’ infection status during the COVID-19 outbreak and their perceived JI. The findings of the study reveal empirical results in understanding employee attitudes toward companies alongside perceived job insecurity for those who are infected as well as for those who are not. Moreover, the study presents theoretical and practical contributions to reduce the negative impact of job insecurity and risk of infection on turnover intentions, which have been considered to be main sources of stress throughout the pandemic.

## 1. Introduction

The world faced a disease named COVID-19 with high infection rates and relatively high mortality rates [1] on 11 March 2020, with the declaration of the WHO [2]. Having quickly become a global scale outbreak, COVID-19 shook the cores of the tourism industry, which was already vulnerable to shocks [3]. Mobility in tourism almost came to a stop due to the pandemic, while the hospitality industry lost significant sales [4,5]. Global tourism mobility decreased by 74% in 2020 in comparison with 2019, while USD 1.3 trillion were lost in global tourism revenue [6]. Bringing tourism and hospitality industries’ activities to a stop, the COVID-19 outbreak also caused a widespread unemployment issue [7,8]. Employees in the hospitality industry faced the risk of losing their jobs, in addition to being exposed to high risks of infection, because they worked in environments that include contact [9]. Having been affected by these negative outcomes, employees in the tourism industry faced intensive stress [10,11] and high levels of job insecurity (JI) [12]. JI was actually one of the most important problems in work life [13]. Still, the negative conditions of the pandemic strained the impact of JI on employees [14,15].

A significant issue in the tourism industry itself, JI increases employees’ turnover intentions (TI) [16,17]. Explained as employees’ consideration of leaving their current jobs and finding another one, TI [18,19] has a positive correlation with actual turnover [20]. Employee turnover is a critical issue for companies as well [21]. That is because there are negative relations between employee turnover and job satisfaction [22], job performance [23], organizational commitment [24], organizational effectiveness [25], and employee productivity [26], and positive relations between employee turnover and human resource cost [27].

Eisenberger et al. [28], within the context of Social Exchange Theory (SET), claim that organizational support (OS) provided by companies to employees would affect employees’ behavior. In straining times such as during crises, OS from companies to employees becomes more important. We believe that providing support to employees in the tourism sector who have been adversely affected by the pandemic, reducing their TI, and increasing their organizational commitment (OC) will provide various advantages. Numerous studies in the literature regarding the pandemic revealed remarkable results such as OS increasing OC [29,30,31], and OS [24,32,33] and OC [34,35,36] decreasing TI. Still, we believe that examining the links between OS, OC, and TI in detail and in relation with employees’ infection status and perceived JI is also very important, because the perceived OS, level of OC, and TI of employees, who were infected during the pandemic, will differ from those who were not infected, despite the direct exposure of both groups to the crisis. In other words, it is estimated here that the effect of OS that is offered during a crisis may differ from the usual effect, and it may also differ by the status of infection of employees. Since hospitality employees had to work in conditions that included direct contact during the pandemic, they are at a higher risk to be infected [37,38]. Thus, we predict that measures to be taken by companies in order to prevent their employees from getting infected, as well as the OS they provide to help employees overcome their status if they are infected, will be stronger than during usual times. Crises such as outbreaks are often destructive events and potentially cause negative attitudes and behaviors in employees toward both their employers and their sectors [39]. The literature review in this study did not yield any studies that examine, at length, the links between OS, OC, and TI within the context of employees’ infection status and perceived JI. Therefore, this study examines the links between OS, OC, and TI within the context of employees’ infection status and perceived JI to fill the existing gap in the literature. In this context, the following moderated mediation research model, as shown in Figure 1, is tested in the study.

Proposed in Figure 1, the conceptual model posits that the indirect impact of perceived organizational support (X) on turnover intentions (Y) mediated by organizational commitment (M) relies on the status of being infected (W) and job insecurity (Z). The model predicts that OS would increase OC (X → M) and OC would increase TI (X → M → Y). It simultaneously suggests that the status of being infected and job insecurity play moderating roles on the indirect effect of OC on TI via OC. We are also responding to the call for research to focus on the welfare of tourism employees [40] with this study. Variables of the study simultaneously contribute in a theoretical sense to SET, since they are based on it. SET is a theory that is often utilized in the literature to explain the relationships between employees and supervisors or employees and organizations [41]. Moreover, many other studies attempt to explain their subject matters within the context of OS [42], OC [43], TI [44], and JI [41]. SET examines the social relationships in the event of an interaction, arising from the changes in material and immaterial resources among individuals and groups [45]. Thus, we are of the opinion that the findings of this study will help us have a better grasp on the impact of OS to be presented during crises on OC and TI within the context of hospitality employees.

### 1.1. The Link between Organizational Support and Turnover Intention

Companies strive to retain their qualified employees to preserve profitability and sustainability in relation with globalization and increasing competition. One of the most important efforts to that end is OS [46,47]. OS is approached by Eisenberger et al. [28] within the scope of SET. Discussing OS within the scope of SET will naturally be of benefit. SET is a general sociological theory concerning the understanding of resource exchange between individuals and groups [48]. SET is often utilized to explain the effects of material and psychological changes on situated attitudes [49]. Blau [50], one of the pioneering names in developing SET, first explicated how social exchange occurs among individuals, then emphasized the ways in which the theory functions between the organization and employee. Blau [50] states that an individual who is good to another will likely feel like they must reciprocate within the scope of social exchange. Even though the time or form is not set, the individual would at least expect to see an act of goodness from the other party. This generates a reciprocal change in necessity and expectation. Such changes can be both emotional and material. According to SET, when organizations’ investments for employee satisfaction are perceived correctly by them, social exchange begins, and the positive relationship continues alongside said perception [51].

SET is often used in the literature to explain the relations between employees and supervisors, or employees and organizations [41]. The concept of OS, developed within the scope of SET, focuses on the link between employees and organization [52]. In other words, OS refers to organizations’ awareness of employees’ contributions to the organization and their needs so that the organization is careful about employees’ welfare [28,53]. OS practices can occur in different forms in organizations. According to Zhang et al. [54], OS can be examined in three categories: work support, personal support, and risk support. Work support refers to the type of support regarding employees’ operational processes, such as providing them with necessary protective gears. Personal support refers to the type for families, such as childcare and family support. Finally, risk support refers to corporate support to protect employees from environmental risks such as delivering timely updates about the pandemic or providing care to employees who are infected.

OS is related with TI, which is another significant variable for organizations. Employees’ TI have been an interesting topic for researchers for some time. In the broadest of terms, TI refers to employees’ considerations and plans of leaving their current jobs and finding other ones soon [18,55,56]. From this perspective, TI can be viewed as a cognitive precursor of quitting behavior [57]. Within the scope of SET, the interaction and relationships between employees and their supervisors and coworkers is likely to make them want to remain in the organization, which would eventually have a negative effect on TI [23,58]. A literature review yields that OS has positive effects on many critical factors for organizations, such as employee turnover rates [24], absenteeism [59], job satisfaction [60], and organizational commitment [30]. In addition, many studies [61,62,63] indicate that TI and absenteeism will decrease in organizations with high levels of OS. Studies conducted in the field of tourism also show that OS has a negative effect on TI. Gök et al. [64] and Akgunduz and Sanli [65] found in their studies in Turkey that OS decreases TI. In another study, conducted with five-star hotel employees in Portugal, it was found that satisfaction in human resources management practices improve OS, which in turn negatively influences TI [66]. Cheng et al. [67] conducted a similar study in Taiwan, where they found that employees’ TI are decreased via OS. Park et al. [68] conducted a study on restaurant employees, where they found that OS increases life satisfaction of employees, which also negatively affects TI. Many studies in this regard can be found in the literature from different countries and in different fields, as they were conducted during the pandemic [24,33,44,69], and they all conclude that OS has a negative effect on TI. Based on these findings in the literature, we propose the following first hypothesis of the study:

**Hypothesis** **1** **(H1).**
*Organizational support has a negative effect on turnover intentions.*


### 1.2. The Link between Organizational Support and Organizational Commitment

OC has been the subject matter of many studies for a long time. OC is defined as an employee’s strong belief in and adoption of organizational goals and values, efforts to sustain their organizational membership, and psychologically committing to the organization [70,71,72]. Relevant literature in this field shows that a number of variables have statistically significant relations with OC, such as burnout [73], emotional labor, job performance [74], job satisfaction [75], job stress, job insecurity [76], turnover intention [34], leadership, productivity, organizational effectiveness [77], organizational citizenship [78], organizational justice [79], and organizational support [30]. Moreover, a number of studies [29,30,31,32,80,81,82,83] concluded that OS improves employee loyalty towards the organization. Bilgin and Demirer [84] and Ersoy [85] both found that OS has a positive effect on OC in the studies they conducted in the Turkish tourism industry. OS is also found to have a positive effect on OC by Kim et al. [86] in the study they conducted in South Korea and by Hemdi [87], Nasurdin et al. [88], and Ramos et al. [82] in Malaysia. He et al. [89] conducted a study in China, where they concluded that dimensions regarding OS have positive effects on OC. Garg and Dhar [52] and Jaiswal and Dhar [90] conducted a study on hotels in India, designating the mediating role of OC in the relationship between OS and service quality. El-Aty and Deraz [91], on the other hand, found in the study they conducted with five-star hotels in Egypt that OS positively affects OC. The literature review shows that this subject is covered in many studies within the context of different countries and fields, all concluding that OS has a positive effect on OC. In this context, we propose the second hypothesis for the study as follows:

**Hypothesis** **2** **(H2).**
*Organizational support has a positive effect on organizational commitment.*


### 1.3. The Link between Organizational Commitment and Turnover Intentions

OC has to do with employees’ attitudes and behaviors toward their job and their organization. High levels of OC naturally reflect positive attitudes and behaviors toward their organization, which is why this term is approached as a variable related to decreasing employees’ likelihood of quitting their jobs [92]. There are many studies in the literature pointing out a negative relationship between OC and TI [34,35,36,93]. Bulşu and Gümüş [94] and Guzeller and Celiker [95] found in their studies, conducted with hotel employees in Turkey, a negative relationship between OC and TI. Lalopa [96] conducted a study on four different hotels in the United States, also finding a negative relationship between OC and TI. Ausar et al. [97] acquired similar results in a study they conducted in the United States with students in the field of tourism, for they are viewed to be potential tourism employees. In another study, conducted in Cyprus, a similarly negative link was identified between OC and TI [98]. In another study, conducted in Indonesia, a negative relationship between OC and TI was identified with the prediction that such relationships undertake the mediating role between job satisfaction and TI [99]. Ampofo and Karatepe [34] identified a partially mediating role between OC’s job embeddedness and TI in the study they conducted with small-scale hotels in Ghana. The literature review shows that this subject matter is analyzed in different countries and in different fields, while the research findings generally indicate that OC has a negative effect on TI. Based on such findings in the literature, we propose the following third hypothesis:

**Hypothesis** **3** **(H3).**
*Organizational commitment negatively affects turnover intentions.*


### 1.4. The Effect of Organizational Support on Turnover Intentions via Organizational Commitment

The literature review shows that OS is related to many different variables. Some of these variables are work commitment [100], employee turnover rate [101], absenteeism [59], organizational citizenship [78], work performance [74], job satisfaction [75], and organizational commitment [30]. In this context and within the scope of this study, OS is predicted to have a positive effect on OC, which can negatively affect TI. Rhoades and Eisenberger [51] revealed in their study that employees’ organizational commitment improves in organizations with high levels of OS, which negatively affects TI. OS has also been proven to be effective on TI via OC by Albalawi et al. [102] in a study which was conducted with SME employees, Nadeem et al. [103] in a study with bank employees, and Xiu et al. [104] with a study with employees at a public university. Chew and Wong [105] also found in their study in Malaysia that OS is both related to OC and TI. In another study, conducted on three-star hotels in Malaysia, managers’ support was found to be effective on employees’ OC and TI [106]. Another study, conducted in the United States with front office employees, found that managerial support has a positive effect on OC, while OC decreases TI [107]. The literature review shows that this subject has been covered in different countries and in different fields. Overall, the findings show meaningful relations among OS, OC, and TI. In this context, the fourth hypothesis of the study is proposed as follows:

**Hypothesis** **4** **(H4).**
*Organizational Support Affects Turnover Intentions via Organizational Commitment.*


### 1.5. The Moderating Role of Infection

The layoff of employees in many countries due to COVID-19 increased unemployment rates and concerns regarding the future [108]. Employees experience both the fear of being let go and the fear of being infected with the disease. While layoffs were prohibited in some countries during the pandemic [109], there was still news about the dismissal of employees who were let go after being diagnosed with COVID-19 [110]. Employees that work with the risk of losing their jobs once they are infected are known to experience higher levels of negative feelings such as high levels of stress and exhaustion than those that do not experience such risks [15]. However, when employees feel supported and valued by their organizations, they can form an emotional bond and start to show more effort for the sake of the organization [28]. According to SET, employee–organization relationships occur through a series of mutual exchanges, even though they are not always simultaneous [50], and policies, adopted by organizations toward their employees in the face of crises, may strengthen or weaken employees’ loyalty. Accordingly, when employees are supported during crises by their organizations, their OCs are stronger [111]. Crises are negative events with the potential to cause destructive stress on employees. When employees who are infected with COVID-19 are not supported by their organization during the crisis and face the risk of being let go, their trust and loyalty toward their organizations will decrease. The exact opposite, on the other hand, would improve their OCs. Galanaki [112], in a study conducted in Greece, found that a decrease in perceived support of employees by organizations during crises affects the level of organizational commitment rather than an increase in perceived benefits. Filimonau et al. [113] conducted a study in Spain with top hotel managers where they found that levels of organizational resilience and corporate social responsibility practices reinforce managers’ perceived job security and improve their OC. Mao et al. [114] conducted a study in the Hubei region of China where they found corporate social responsibility practices of organizations improve respondents’ levels of satisfaction toward organizations’ responses to the COVID-19 crisis, in addition to making them more hopeful and optimistic. Organizations’ support to their employees during crises can be said to increase the feeling of gratitude within the context of SET when it comes to employees, revealing their loyalty to organizations. Furthermore, we are of the opinion that employees’ commitment toward their organization throughout COVID-19 will differ as per their status of infection and the support they received from their organizations during this time. In light of these explanations, we propose the following hypothesis:

**Hypothesis** **5** **(H5).**
*The Effect of Organizational Support on Organizational Commitment Differs by the Infection Status of Employees.*


### 1.6. The Moderating Role of Job Insecurity

Data regarding tourism employment show that no other crisis has impacted the world tourism industry as much as COVID-19 did [115]. The COVID-19 outbreak generated a shock in employment across the world, in addition to making the industry come to a halt. Accor Group shut down two thirds of their hotels during COVID-19. Marriott International put almost 174,000 employees from all levels on unpaid time off and paid nothing to their employees during this time except health support [116]. In the USA, 70% of those working in the hospitality and restaurant industries were either laid off or furloughed due to the COVID-19 outbreak [39]. Despite all the precautions taken, COVID-19 continued to spread, which enhanced the ambiguous unemployment of tourism employees [117] and perceived JI.

JI is defined in the broadest of terms as an employee’s worries about the future of their job [118]. Heaney et al. [119] approach JI as the perception of a potential threat toward an employee’s continuation of their current job. In other words, JI arises in cases where the sustainability of a job cannot be estimated [120]. This study approaches JI as the consequence of a health crisis, which arose due to COVID-19 pandemic, which affected the tourism industry across the world. JI also encompasses the difficulty and ambiguity of finding a new job if the current job is lost [121]. In line with this result, a study conducted by Baert et al. [122] found that 52% of the respondents in the study believe COVID-19 can negatively affect their work prospects. In this context, job potentials and unemployment rates in the sector also affect employees’ TI regarding their current jobs [123]. The COVID-19 outbreak led to a number of destructive effects on all tourism companies, in addition to hotels, which limited alternative job opportunities in the field. Naturally, it is predicted that perceived JI of employees without any experience or competence outside the tourism industry will limit alternative job opportunities. Job alternatives include opportunities that can be easily found by employees if they leave their jobs and would be willing to work [124]. When alternative job opportunities become abundant, employees’ quitting behavior in relation with their skills also increases, and organizations’ turnover rates do as well [125,126]. However, even though perceived JI of employees in the tourism sector increased during the pandemic, their TIs had to decrease, because no better alternatives were present under the circumstances of the job market.

While TI is an indicator of low OC, it may not always turn into behavior. Yıldırım et al. [100] found in a study they conducted in Turkey that OC has a negative but low-level effect on TI. Uludağ et al. [127], on the other hand, did not find any significant links between OC and TI. Employees’ perception of alternative job opportunities takes up significant space in understanding TI. In other words, the effect of OC on TI can differ as per alternative job opportunities. The scarcity of job alternatives can reduce the effect of OC on TI or may vanish it completely [57]. In this context, we predict that TI of employees with low OC will increase in circumstances under which they perceive that alternative job opportunities are more likely to be found, whereas TI will remain weaker when they are less likely to find a job. Therefore, the following hypotheses are posited:

**Hypothesis** **6** **(H6).**
*The effect of employees’ organizational commitment on their turnover intentions differs by their perceived job insecurity.*


**Hypothesis** **7** **(H7).**
*The status of being infected and job insecurity play moderating roles on the indirect effect of perceived organizational support on turnover intentions via organizational commitment.*


## 2. Materials and Methods

### 2.1. Participants and Procedure

This study was conducted with five-star hotel employees in Alanya, Turkey. The research employs a quantitative method and is cross-sectional. Employees with at least 3 months of experience at their organization comprise the potential respondents of the study. To ensure that each respondent was fit for the study, they were asked how long they have been working at their organization in the questionnaire. Employees who worked at their organization for less than 3 months were excluded from the study. We also added two attention checks to the questionnaire (1. I filled the questionnaire without reading the questions, 2. I answered all questions incorrectly). Questionnaires of respondents who agreed with these questions were excluded. Data from the study were collected in August–September 2021—during the busy tourism season in Alanya. The study was conducted under the prevailing conditions of COVID-19, which is why convenience sampling was adopted to collect data from twenty-three accommodation companies that are accessible. Human resources managers at the organizations that agreed to take part in the study were given a presentation about the study’s aim, importance, and methods, and were provided with questionnaires. They were later distributed to others in the department by human resources and were later collected. Respondents were delivered questionnaires in envelopes so that they can easily fill them without having to fear hotel management. Moreover, questionnaires also included information describing the purpose of the study, indicating that participation was voluntary, and informing that the data collected were to remain completely confidential. Respondents were asked to hand in their questionnaires in closed envelopes. Hotels were revisited 15 days later and the questionnaires were collected. A total of thirty-eight questionnaires were excluded because they were not put into closed envelopes or because the envelopes remained open. In total, 517 questionnaires were acquired from twenty-three five-star hotels. However, after the elimination of the incorrectly answered 27 questionnaires (missing & outliers), we analyzed 490 questionnaires. As recommended by Mohseni et al. [128], we used an online a priori sample size calculator to determine the sample size for our presumptive model [129]. From this framework, the number of observed (21) latent constructs (4), anticipated effect size (0.25), probability level (0.05), and desired statistical power level (0.95) were calculated to determine sample size. Consequently, minimum sample size to test our research model was calculated to be 316. According to this calculation, a sufficient sample size was met for the study. Having been conducted with quantitative research methods, this study collected data via questionnaires. On account of social isolation and social distance rules, research data were collected with convenience and snowball sampling methods in order to avoid close contact with third parties. Due to the challenges in collecting data under the prevailing conditions of the pandemic, the research study is cross-sectional and uses the convenience sampling method from the nonprobability sampling methods. The researchers strictly followed ethical principles (e.g., respect, autonomy, confidentiality, beneficence, and nonmaleficence) to ensure research integrity. All respondents gave their informed consent. To encourage participation, respondents were told in oral and written forms that they can leave any time they wish to do so without any justifications at all. No identifiable information was asked in the questionnaire for anonymity purposes. No experimental and clinical data were collected from the participants of the study.

### 2.2. Measurements

A self-administered survey was used in this study to identify the moderator role of infection status and job insecurity on the effect of hospitality employees perceived organizational support on their turnover intentions as mediated by organizational commitment. Scales, the validity of which were provided in previous studies, were used to measure the constructs in the research model in Figure 1. Perceived job insecurity of employees was measured with a four-item job insecurity scale (JIS) developed by De Witte [130] and verified by Vander Elst et al. [131]. The scale has a single factor structure, and a high score from it indicates that the employee experiences high levels of job insecurity. Perceived organizational support of employees was measured with the Perceived Organizational Support Scale (POSS), developed by Eisenberger et al. [132], and consisted of eight propositions, which are frequently encountered in other studies as well. Statements in the POSS express what employees think about the extent to which the organization values employee contributions and cares about their welfare. Organizational commitment of employees was measured with a six-item revised [133] version of the emotional commitment scale, developed by Allen and Meyer [70]. Turnover intention of employees was measured by a three-item scale, adapted by Singh and Srivastava [134]. A high score from the scale indicates a high level of turnover intention. All items were measured using a seven-point Likert scale (from 1 = strongly disagree to 7 = strongly agree). In addition, a personal information form was used to collect data regarding respondents’ age, gender, level of education, department of employment, and infection status. Gender, marital status, education, and tenure were used as controlling variables in the study. High scores express more educated employees with longer durations of employment. Gender (0 = female and 1 = male) and marital status (0 = single and 1 = married) are coded as binary variables.

### 2.3. Data Analysis

Data were analyzed with the help of SPSS 25.0 (Windows, IBM Corp., Armonk, NY, USA) and AMOS 24.0 packages (SPSS Inc., Chicago, IL, USA). Data were first screened for missing values and outliers. Consequently, eighteen questionnaires were excluded from evaluation because they revealed missing data above 5%. Mahalanobis’ distance was checked for outlier values. As a result, nine questionnaires were excluded from evaluation. In total, twenty-seven questionnaires were excluded from the analysis due to missing and outlier values. After the data screening, normal distribution assumption was checked with respect to collinearity. Normality assumption was tested by calculating skewness and kurtosis values. Additionally, tolerance and variance inflation factor (VIF) were calculated to determine whether or not multicollinearity existed [135]. Using Harman’s single factor test, common method variance (CMV) was tested [136]. Later, the fit, reliability, and validity of the research model was evaluated with confirmatory factor analysis (CFA). As for the structural validity of the scales, model fit and convergent and discriminant validities were calculated. At the same time, the research model was examined in comparison with alternative models for the fit. SPSS PROCESS Macro was used, as recommended by Hayes [137], to test the posited hypotheses.

## 3. Results

### 3.1. Demographic Findings

Demographics of the respondents are presented in Table 1. In total, 60% of the respondents are male and 40% are female from amongst 490 respondents, and 70% of the respondents are between the ages of 18 and 33, while 59% are married. Furthermore, 9.4% of respondents had been working at their organization for less than a year; 16.9% for 1–3 years; 30% for 4–6 years; 21.6% for 7–9 years; and 22% for 10 years or more. As to the level of education, the majority of respondents (39.0%) are high school graduates. Additionally, 24.7% are primary school graduates, 19.4% are college graduates, and 16.9% hold undergraduate or higher degrees. Finally, 18.6% of the respondents reported that they were infected with COVID-19, while 81.4% did not.

### 3.2. Results of Measurement Model Assessment

CFA was conducted with AMOS 24 to test the statistical fit of the research model. According to the CFA results in Table 2, all factor loads of scale items are above 0.50 and significant at *p* < 0.001 [138]. The presumptive research model was also compared with alternative models with χ^2^ difference tests. The presumptive research model was seen as the model with the best fit, as can be seen in Table 2 (χ^2^ = 386.13, df = 181, χ^2^/df = 2.13, *p* < 0.001, RMSEA = 0.048, SRMR = 0.035, CFI = 0.975, IFI = 0.975, NFI = 0.954, RFI = 0.947) [139]. To assess the normality of the data distribution, skewness and kurtosis values were calculated. According to the results in Table 2, the skewness values are between −0.07 and 0.54; the kurtosis values are between −0.21 and −0.86. Accordingly, data for our research model show normal distribution [140]. For multicollinearity, tolerance and VIF values were calculated. Tolerance values are between 0.549 and 0.952; VIF values are between 1.050 and 1.823. These results indicate that there is no multicollinearity issue [141]. To test CVM, Harman’s single-factor test was applied [136]. The analysis showed that variance explained with a single factor (35.48%) is lower than the cut-off value of 50. Accordingly, there is no CVM. Goodness-of-fit values regarding the models are presented in Table 3. In addition to structural validity for the scales, convergent and discriminant validities were also examined. Calculated for each scale’s convergent and discriminant validities, CR, AVE, MSV, and ASV values are provided in Table 4. AVE values for each scale are greater than 0.50; CR values are greater than 0.70 and CR values are greater than AVE values, all of which indicate convergent validity for the scale [142]. AVE values are greater than MSV and ASV values for scales, while the factors’ AVE square root values are greater than the correlation coefficient between factors, which indicates discriminant validity for the scales [143]. Moreover, Cronbach’s alpha values for each scale is above 0.70, which shows internal consistency [144]. These findings show that scales used in the study have convergent and discriminant validities with high levels of internal consistency. According to the correlation analysis, the results of which can be found in Table 4, statistically significant relations are observed between variables. One of the control variables, tenure, is positively linked with POS (r = 0.17, *p* < 0.01) and CO (r = 0.13, *p* < 0.01). There is a positive link between gender and POS (r = 0.10, *p* < 0.05). There is a positive relationship between marital status and JI (r = 0.13, *p* < 0.01). These positive correlations show that as tenure at an organization increases, perceived organizational support and organizational commitment also increase. At the same time, married employees experience higher levels of job insecurity. According to the correlation coefficients in Table 4, POS positively related with OC (r = 0.66, *p* < 0.01). On the other hand, POS has a negative relationship with JI (r = −0.26, *p* < 0.01) and TI (r = −0.15, *p* < 0.01). OC is slightly negatively correlated with JI (r = −0.15, *p* < 0.01) and TI (r = −0.23, *p* < 0.01). JI has a low positive correlation with TI (r = 0.11, *p* < 0.05).

### 3.3. Hypothesis Testing

Regression analysis based on the bootstrap method was used to test the research hypotheses. Analyses were conducted with Model 1, Model 4, and Model 21 from amongst the process macros as developed by Hayes [137]. A total of 5000 resampling options were selected with the bootstrap technique for analyses. In regression analyses conducted with bootstrapping, a 95% confidence interval (CI) levels should exclude zero (0) values acquired as a result of the analysis to support the hypotheses [137]. Regression analyses for the research hypotheses are provided in Table 5. First, the H1 (POS → TI) hypothesis was tested. According to regression analysis results in Table 5 Model 1, POS negatively affects TI (β = −0.15, 95% CI [−0.23;−0.06], t = 3.42, *p* < 0.001). Furthermore, control variables are observed to have no significant effects on TI. According to these results, the H1 hypothesis is supported. The second hypothesis of the study (POS → OC) was tested with the results indicating that POS has a significant and positive effect on OC (β = 0.66, 95% CI [0.59; 0.73], t = 19.03, *p* < 0.001). At the same time, marital status, which is one of the control variables, positively predicts OC (β = 0.0, 95% CI [0.02; 0.42], t = 2.13, *p* < 0.05). Accordingly, the H2 hypothesis is supported. The results of the regression analysis where OC is the mediation variable found that OC has a negative effect on TI (OC → TI) (β = −0.19, 95% CI [−0.30; −0.08], t = −3.25, *p* < 0.001). This result supports the H3 hypothesis. Additionally, POS is determined to have an indirect and significant effect on TI as mediated by OC. The indirect effect of POS on TI is statistically significant (POS → OC → TI) (β = −0.12 95% BCA CI [−0.20; −0.05]). The inclusion of OC as a mediator variable caused the effect of POS on TI to lose its significance (direct effect of POS on TI: β = −0.02 95% BCA CI [−0.13; 0.09]). This finding shows that OC has a fully mediating role on the relation between POS and TI. This result supports the H4 hypothesis.

The effect of POS on OC is predicted to differ by the infection status of employees throughout the COVID-19 pandemic. In this context, the effect of organizational support on organizational commitment was tested with respect to the moderator role of employees’ infection status. According to the results of the regression analysis in Table 5 Model 2, POS had a positive and significant effect on OC (β = 0.57, 95% CI [0.50; 0.64], t = 15.83, *p* < 0.001), while being infected or not did not yield any significant effect (β = −0.07, 95% CI [−0.29; 0.16], t = −0.56, *p* > 0.05). However, POS and the status of infection were detected to have a significant interaction effect on OC (β = −0.45, 95% CI [−0.58; −0.32], t = −6.79, *p* < 0.001). Marital status coefficient was significant and positive, indicating higher levels of OC than single respondents. Accordingly, the effect of POS on OC differs by the status of infection.

A moderator variable helps to understand the cases in which the relationship between two variables increases, decreases, or changes direction. When the moderator effect is analyzed at length from this perspective, the effect of POS on OC appears to be strong for infected employees (β = 0.92, 95% CI [0.83; 1.03], t = 17.91, *p* < 0.001), while for noninfected employees, the effect is at a medium level (β = 0.47, 95% CI [0.40; 0.56], t = 11.52, *p* < 0.001). Consequent to the slope analysis, the effects of moderator variables are provided in Figure 2. These results support the H5 hypothesis.

Table 5 Model 3 shows the results of the regression analysis, displaying JI’s moderator role. According to the results of the analysis, OC negatively affects TI (β = −0.17, 95% CI [−0.29; −0.09], t = −4.17, *p* < 0.001), and JI positively affects it (β = 0.09, 95% CI [0.01; 0.16], t = 2.24, *p* < 0.05). The OC and JI variables are also determined to have significant interaction effects on TI (β = 0.11, 95% CI [0.06; 0.15], t = 4.91, *p* < 0.001). Accordingly, the effect of OC on TI differs as per employees’ perceived JI. The details of the moderating effect show that OC has a significant and negative effect on TI when perceived JI is high (β = −0.38, 95% CI [−0.49; −0.27], t = 6.63, *p* < 0.001). Still, when perceived JI is low, the effect of OC on TI is not significant (β = −0.01, 95% CI [−0.11; −0.10], t = 6.63, *p* > 0.05). Figure 3 shows the effect of OC on TI as per situations of JI. Consequently, the effect of OC increases on TI when perceived JI is high for employees. These findings support the H6 hypothesis.

The moderated mediation role of POS, in other words, whether or not its indirect effect on TI as mediated by OC is linked with being infected and JI, is also tested. As shown in Table 5 Model 4, the index of the moderated mediation value is significant (β = −0.05, 95% CI [−0.08; 0.03]), as the variables of infection status and JI play moderator roles on the indirect effect of POS on TI via OC. According to the results in Table 5 Model 4, employees who were infected during the COVID−19 pandemic with high levels of perceived JI display the strongest levels of the indirect effect of POS on TI via OC (β = −0.37, 95% CI [−0.50; −0.24]). Employees who are not infected and did not experience JI did not reveal significant effect of POS on TI via OC. These findings support the H7 hypothesis.

## 4. Discussion

An unprecedented situation, namely, the outbreak of COVID-19 [145], had quite a negative influence on the psychological well-being of people [146]. Throughout the outbreak, individuals had to struggle with myriad psychological problems such as fear, panic attacks, depression [147], being upset, frustration [148], anxiety, insomnia, stress [149], post-traumatic stress [150], eating disorders, and suicidal tendencies [151]. These problems were experienced more severely by tourism employees, whose risk of contracting COVID-19 were almost as high as that of health workers [152,153]. For employees in the tourism sector, challenges they had to face during COVID-19 were not limited only to psychological problems [154]. During this time, employees in the tourism sector, who had to fight many psychological challenges [155], also experienced sharp declines in their incomes [114] and had to look for other job opportunities due to being put on unpaid leaves [156,157]. These conditions lowered employees’ OC [113] and increased their TI [17]. Employees had to face high levels of JI during the pandemic due to such challenges [158]. In these cases, employees naturally need more OS [15,159]. This study examines the relations between OS, OC, and TI of employees in the hospitality industry during the COVID-19 pandemic with respect to their infection status and perceived JI. Accordingly, the important findings are presented below.

First, we determined that OS has a negative effect on TI in the analyses, carried out in line with the hypotheses of the study. This finding is in accordance with those of Asghar et al. [24] in Pakistan; Guna and Satrya [32] in Bali, Indonesia; Jolly et al. [33] in the USA; and Raza et al. [44] in Lahor, Pakistan, where hospitality employees were investigated as well. To acquire the expected results from OS, support must be offered correctly and at the right time [69]. In this context, the findings of these studies, which were conducted with hospitality employees during the pandemic, reveal that OS presented during this time decreases TI. Furthermore, Self et al. [160] found in a study they conducted with restaurant managers in the USA that a relationship similar to the negative one between OS and TI exists between coworker support and TI. All these findings show that the ties between organization and employee are reinforced and employees’ TI decreases if they think the OS they are provided is sufficient. OS plays a significant role for the hospitality industry, “having increased the employees’ trust level, improving the organization’s beliefs, contributions, and care support to employees’ well-being and job engagement” [24]. 

Secondly, we have determined that OS positively affects OC in the study. This finding is correlated with those of Kim et al. [30] regarding airline employees; Suwandana et al. [31] in Indonesia; Bae [29] in the USA; Guna and Satrya [32] in Bali, Indonesia; Ramos et al. [82] in Malaysia; and Salem et al. [83] with Egyptian hotel employees. The findings of these studies, which were conducted during the pandemic in different countries with different samples, show that OS provided during this time positively affects employees’ levels of OC. Referring to employees’ feelings of loyalty for their organizations of employment, OC [70] is closely related with OS [161], because employees’ OC can be improved with the organization’s OS behavior [162]. Moreover, this relationship can be explained within the scope of SET. If employees witness supportive behavior from the organization such as human resource practices, rewards, or procedural justice, such exchange leads to the emergence of loyalty, according to SET [161,163].

Thirdly, another finding of the study was that OC negatively affects TI. Porter et al. [164] stated that employees will remain in the organization as long as their OC levels are high. This evaluation is still valid as we can see. Findings of the studies carried out during the pandemic also point out that employees’ levels of TI decrease as their levels of OC increase. The findings of the studies by Ampofo and Karatepe [34] in Ghana, Tsaousoglou et al. [36] in Greece, Murray and Holmes [35] in Canada, and Yan et al. [93] in Chinese hotel employees correlate with the findings of this study. OC is a variable that is intricately linked with TI [165]. The lack of developing any sort of commitment toward the organization would negatively affect employees’ intentions to leave their current organizations [166]. Various studies in the literature [17,167] state that practices that would improve employee OC to decrease their TI are correct.

Fourth, the study found that OC plays a fully mediator role on the effect of OS on TI, which is in accordance with prior studies in the literature [102,103,104,168]. According to the findings of this study, organizations’ support for their employees during challenging times both positively impacts OC and decreases their TI as a result of such interaction. However, Saralita and Ardiyanti [169] found in a study they conducted with the employees of a private hospital in Jakarta, Indonesia that OC does not have a significant mediator role on the effect of OS on TI.

Fifth, the findings indicate that if the perceived JI of employees is high, then the effect of OC on TI is significant and negative. Still, when perceived JI is low, the effect of OC on TI is not significant. Under normal circumstances, employees’ TI increases alongside employee turnover rates, when alternative job opportunities are abundant in the industry [125,126]. In other words, alternative potential jobs in the market affect employees’ TI concerning their current jobs [123]. However, the COVID-19 outbreak significantly lowered tourism employees’ access to alternative job opportunities [17,114,156]. At the same time, the outbreak had a negative impact on employees’ perceptions of JI [12,15,38,158]. Naturally, employees in the tourism industry simultaneously started to worry about losing their jobs and witness the lack of alternative job opportunities or limited opportunities, which made the effect of OC on TI more significant. Based on this finding, practices to improve OC during unusual times of crises such as during a pandemic, when perceived JI of employees become specifically high, can significantly decrease TI. In other words, if organizations display OS behavior that makes employees feel like their organizations value them during times of crises, when they most intensively feel the fear of losing their jobs, their OC can be strengthened more than during usual circumstances, which decreases TI, according to our findings. 

Finally, our study found that the effect of perceived OS of employees who were infected, on OC, is quite powerful, whereas for employees who were not infected, the effect is at a medium level. The indirect effect of OS on TI as mediated by OC in employees who were infected during the pandemic and have high levels of perceived JI reaches its peak. These findings show that OS improves the OC of employees who were infected with COVID-19. The pandemic caused tourism industry employees to have higher infectious risk perceptions, job-stress levels, perceived JI, and TI levels [14,15], simultaneously rendering their working conditions more challenging. If organizations provide sufficient and correct OS to their employees, in addition to taking measures in ways to increase their organizational resilience, the increasing OC is likely to decrease their TI [34,36,113]. According to Zhang et al. [54], risk support refers to providing corporate support to protect employees from the effects of COVID-19 such as giving updated information about the pandemic or ensuring care for infected employees. From this perspective, we can say that providing risk support to employees during the pandemic increases employees’ OC—especially for the ones that are infected. Moreover, we are of the opinion that this finding has theoretical contributions for SET. According to Eisenberger et al. [170], employees feel like they owe the organization as a result of the positive behavior they receive there and wish to reciprocate. Sungu et al. [171] found in a study they conducted that perceived OS improves employees’ levels of OC. Researchers have pointed out that this finding is a reaction by employees in the face of positive behavior they encountered. We believe that this evaluation is in direct correlation with the findings of this study. Therefore, we argue that within the context of SET, the positive support infected employees will receive from their organization will help the organization to be reflected upon in a much more positive light.

### 4.1. Theoretical Implications

The findings of this study present various theoretical implications. First, there are many studies that investigate the links between OS, OC, and TI [102,172,173,174]. However, no study that investigates such relationships within the context of perceived JI and infection status of employees in the tourism sector during the pandemic was found. Therefore, the findings of this study provide specific contributions to the theoretical field by presenting detailed outcomes as per the level of being affected by the crisis. Secondly, the positive effect of OS on OC and its negative effect of TI has been revealed in prior studies [24,29,30,31,32,33], which presented results similar to the findings of this study. In addition, Wang and Wang [69] stated that the effect of OS will be much more powerful when it is provided at the right time. The finding we acquired in this study to that end was that OS generates a strong effect in employees when it is offered in times of crises such as during a pandemic, which reinforces the theoretical evaluations in the field. Third, increasing perceived JI of employees often improves their TI tendencies [16,17], which leads to actual turnover outcomes [20]. Still, the findings of this study indicate that the challenges of having access to alternative job opportunities during the pandemic makes the effect of JI on TI insignificant for employees in the tourism sector. This finding yields a different approach in theory than other findings in the literature because the positive relationship between JI and TI appear to occur differently in crises than they do under normal circumstances. Fourth, SET argues that positive OS provided to employees affects employee behavior [28], and employees wish to reciprocate the positive behavior they observe in the same way [170]. The findings of this study showed that the effect of perceived OS on OC is quite powerful for individuals who are infected. Moreover, for infected individuals with high levels of perceived JI, the indirect effect of OS on TI as mediated by OC reaches its strongest level. These findings are definitely in accordance with the tenets of SET, in addition to making specific contributions to it.

### 4.2. Practical Implications

While some think that COVID-19 will vanish for good soon, it is still impossible to say anything definite for the time being. Organizations must continue to provide risk support to their employees and consider it a strategic action of human resources to increase employees’ OC and decrease their TI. According to Galanaki [112], a decrease in perceived support from the organization during times of crises affects employees’ levels of organizational commitment more than any increases in perceived perks and benefits.

We predict that tourism employees will have access to myriad alternative job options following the pandemic’s aftermath in near future, which will significantly improve the TI of employees who perceive the currently provided OS to be insufficient, aside from having low levels of OC and high levels of JI. Thus, organizations must immediately take measures during this time to decrease JI and increase OC of their employees in the form of yet another strategic human resources action. Making such investments can convince employees to remain at their organizations instead of turning to alternative opportunities. This kind of a step can actually reduce the losses in human resources that organizations and the sector itself may suffer after the pandemic. During the pandemic, many skilled tourism professionals in Turkey had to switch to other jobs, such as shipping and courier services, because they felt they did not receive sufficient support from their organizations. Thus, it has become difficult to find qualified personnel in the sector. It has even been discussed that if this change in jobs is permanent for these qualified workers, hotel owners would have to change their fields of work too [175,176]. In fact, this is a recent and drastic experience, revealing the critical outcomes of necessary support that must be provided by organizations to their staff during times of crises. Thus, researchers, public authorities, nongovernmental organizations, and unions in tourism must develop a system to prevent employees from giving up on the tourism sector in future crises. We believe that such a step possesses strategic importance on a macrolevel for the sector. 

### 4.3. Limitations and Future Research

Even though remarkable findings for the tourism sector were acquired in this study, there are still limitations to it. Despite the positive correlation between TI and actual turnover [20], only time can show whether or not such intentions will be converted into behavior. Therefore, TI as observed in this study shall not be perceived as definite proof of actual turnover. In this context, we believe that it is important to investigate the matter of actual turnover in future studies. The study approaches OS in general. However, future studies can probe this concept within the scope of different supporting actors such as manager support or coworker support. Such examinations can help researchers have access to detailed findings concerning the effects of OS on OC and TI. Due to the challenges in collecting data under the prevailing conditions of the pandemic, the research study is cross-sectional and uses the convenience sampling method from the nonprobability sampling methods. Convenience sampling is easy, economical [177], and often preferred in hospitality sector studies. Still, this method makes it difficult to generalize the findings. Therefore, utilizing probability sampling methods to collect data in future studies may bring along certain advantages in terms of coming to generalizable conclusions. In the first period of the pandemic process in Turkey, the number of employees in the hospitality sector decreased by 39.7%. The level of employment in the hospitality sector was more affected by the pandemic than in other sectors [178]. The COVID-19 pandemic has also severely limited alternative job opportunities for tourism industry employees. As a matter of fact, this situation may affect the perception of JI more negatively by employees who do not have experience or competency outside the hotel industry. Most of the participants in the study were quite young, had a relatively low level of education, and at the same time, about half were married. This may be a combination of factors that influence a person’s anxiety about a possible job loss. Data for this study were collected with the help of employees in the tourism sector in Alanya, which is one of the top tourist destinations in Turkey. Therefore, the findings may not provide any ideas about tourism employees in other tourist destinations in Turkey or across the world. Collecting data in more extensive ways in future studies will help to make stronger assessments. Data for this study were analyzed with quantitative methods. Using qualitative methods and a longitudinal research design in future studies can also contribute to reaching more comprehensive conclusions.

## 5. Conclusions

The results obtained in this research revealed the important factors affecting the turnover intention of hospitality sector employees in times of crisis such as during epidemics. The study concluded that organizational support (OS) and commitment (OC) have reducing effects on employees’ turnover intention (TI), and these effects differ significantly according to employees’ perceptions of being infected and job insecurity (JI). In more detail, it was concluded that the effect of OS on OC differs by employees’ status of being infected. This finding reveals that especially in difficult times such as crises, the support provided by organizations to employees strengthens the commitment of the employees more than during usual times. In other words, when provided by organizations for their employees, organizational support generates a stronger and more positive impact on employees, who are directly affected by crises such as pandemics. Another important finding obtained in the study is that the effect of OC on TI differed significantly according to employees’ perceptions of JI. The findings indicate that if the perceived JI of employees is high, then the effect of OC on TI is significant and negative. When perceived JI is low, the effect of OC on TI is not significant. The findings revealed that, as perceived JI of employees decreases, OC loses its effect on TI. This result revealed that conditions with limited alternative job opportunities significantly increased the impact of OC on TI. This means that the difficulty in accessing alternative job opportunities during the COVID-19 pandemic makes the otherwise positive effect of job insecurity on turnover intentions insignificant. Overall, it was determined that the indirect effect of OS on TI as mediated by OC in employees, who were infected during the pandemic and had high levels of perceived JI, was strongest. The results obtained in the research provide theoretical and practical contributions to reducing the negative impact of job insecurity and risk of infection on turnover intentions, which have been considered main sources of stress throughout the pandemic.

## Figures and Tables

**Figure 1 ijerph-19-08516-f001:**
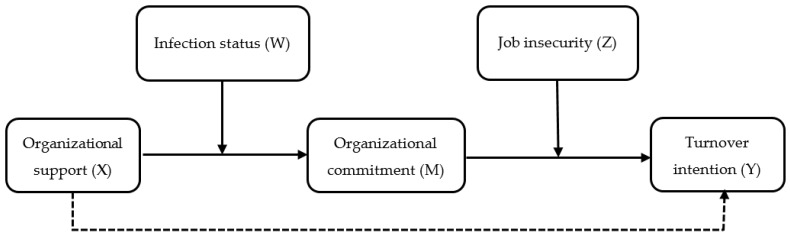
Conceptual model.

**Figure 2 ijerph-19-08516-f002:**
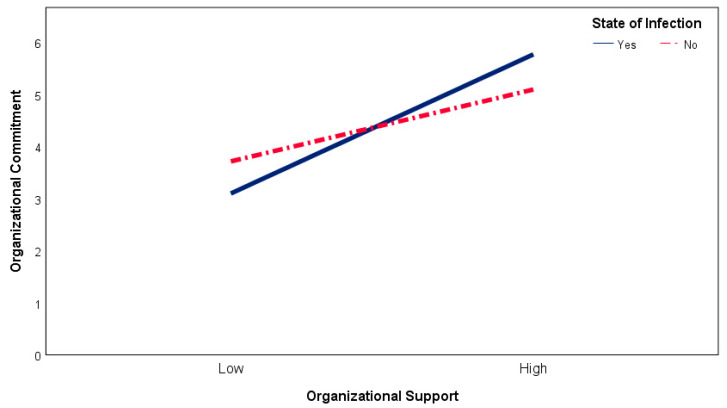
Moderating role of state of infection on organizational support–organizational commitment relationship.

**Figure 3 ijerph-19-08516-f003:**
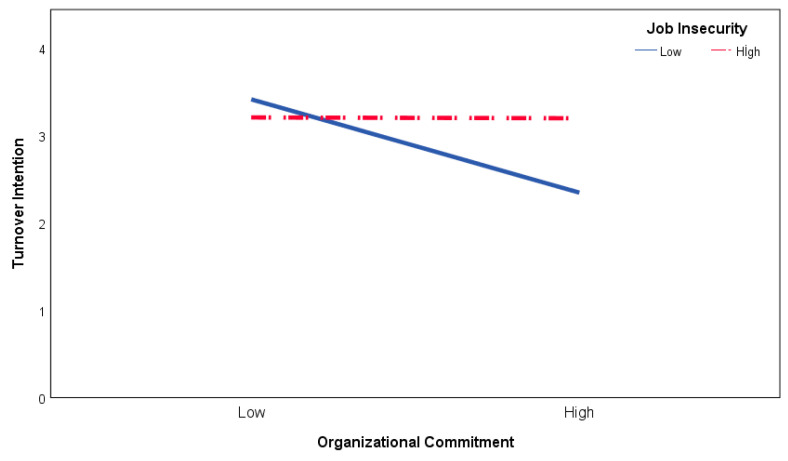
Moderating role of job insecurity on organizational commitment–turnover intention relationship.

**Table 1 ijerph-19-08516-t001:** Sociodemographic details of respondents (*n* = 490).

Characteristics	*n*	%	Characteristics	*n*	%
Gender			Marital status		
Female	196	40.0	Married	289	59.0
Male	294	60.0	Single	201	41.0
Age			Education		
18–25	192	39.2	Primary school	121	24.7
26–33	154	31.4	High school	191	39.0
34–41	98	20.0	College	95	19.4
≥42	46	9.4	Bachelors’and upper degrees	83	16.9
Tenure (year)			COVID-19 infection status		
<1	46	9.4	Yes	91	18.6
1–3	83	16.9	No	399	81.4
4–6	147	30.0			
7–9	106	21.6			
≥10	108	22.0			

**Table 2 ijerph-19-08516-t002:** Measurement model evaluation.

	Mean	Estimate	S.E.	t	Skewness	Kurtosis
**Perceived Organizational Support**						
My organization cares about my opinions.	4.31	0.84	Fixed		−0.29	−0.82
My organization really cares about my well-being.	4.29	0.83	0.03	26.57 ***	−0.10	−0.49
My organization strongly considers my goals and values.	4.48	0.82	0.04	22.74 ***	−0.19	−0.84
Help is available from my organization when I have a problem.	4.46	0.81	0.04	22.08 ***	−0.22	−0.61
My organization would forgive an honest mistake on my part.	4.36	0.88	0.04	25.60 ***	−0.17	−0.78
If given the opportunity, my organization would take advantage of me.	4.32	0.83	0.04	22.88 ***	−0.19	−0.71
My organization shows very little concern for me.	4.42	0.87	0.04	24.65 ***	−0.07	−0.75
My organization is willing to help me if I need a special favor.	4.45	0.83	0.04	22.57 ***	−0.14	−0.79
**Organizational Commitment**						
I would be very happy to spend the rest of my career with this organization.	4.44	0.82	Fixed		−0.27	−0.77
I feel a strong sense of belonging to my organization.	4.49	0.86	0.04	22.87 ***	−0.23	−0.47
I feel like part of the family at my organization.	4.53	0.84	0.05	22.07 ***	−0.15	−0.83
I feel emotionally attached to this organization.	4.40	0.85	0.04	22.47 ***	−0.07	−0.69
This organization has a great deal of personal meaning for me.	4.34	0.87	0.04	23.07 ***	−0.12	−0.67
I really feel as if this organization’s problems my own.	4.51	0.77	0.05	19.31 ***	−0.13	−0.86
**Job Insecurity**						
Chances are, I will soon lose my job.	4.40	0.80	Fixed		−0.51	−0.56
I am afraid that I may not be able to keep my job.	4.55	0.82	0.04	25.28 ***	−0.44	−0.35
I feel insecure about the future of my job.	4.52	0.89	0.06	21.38 ***	−0.42	−0.66
I think I might lose my job in the near future.	4.52	0.87	0.05	21.03 ***	−0.46	−0.70
**Turnover Intention**						
I often think about quitting my job.	3.12	0.82	Fixed		0.38	−0.29
I am actively searching for an alternative to my present job.	3.04	0.87	0.05	20.23 ***	0.48	−0.31
As soon as possible, I will leave my company.	3.00	0.82	0.05	19.64 ***	0.54	−0.21

*** *p* < 0.001.

**Table 3 ijerph-19-08516-t003:** Goodness-of-fit values regarding the models.

Models	X^2^	df	X^2^/df	CFI	RMSEA		Model Comparison		
							∆X^2^	∆df	*p* (∆X^2^)
1. Hypothesized model ^a^	386.13	181	2.13	0.975	0.048		-	-	
2. Three-factor model ^b^	1144.2	184	7.85	0.848	0.118	2 vs. 1	758.06	3	0.000
3. Two-factor model ^c^	2809.2	187	15.02	0.683	0.169	3 vs. 1	2423.09	6	0.000
4. One-factor model ^d^	3542.3	188	18.84	0.594	0.191	4 vs. 1	3156.19	7	0.000

^a^ = Perceived Organizational Support; Organizational Commitment; Job Insecurity; Turnover Intention; ^b^ = Perceived Organizational Support + Organizational Commitment; Job Insecurity; Turnover Intention; ^c^ = Perceived Organizational Support + Organizational Commitment + Job Insecurity + Turnover Intention; ^d^ = Perceived Organizational Support + Organizational Commitment + Job Insecurity + Turnover Intention.

**Table 4 ijerph-19-08516-t004:** Correlations, convergent, and discriminant validity.

Variables	Mean	SD	POS	OC	JI	TI	α	AVE	CR	MSV	ASV
GEN	1.60	0.49	0.10 *	0.06	0.01	0.05	-	-	-	-	-
MAR	1.41	0.49	−0.01	0.06	0.13 **	−0.05	-	-	-	-	-
EDU	2.58	1.01	−0.03	−0.06	0.01	−0.04	-	-	-	-	-
TEN	3.30	1.24	0.17 **	0.13 **	−0.01	0.03	-	-	-	-	-
POS	4.69	1.40	[0.88]				0.95	0.70	0.95	0.44	0.18
OC	4.45	1.39	0.66 **	[0.84]			0.93	0.70	0.93	0.44	0.17
JI	4.50	1.53	−0.26 **	−0.15 **	[0.85]		0.92	0.72	0.91	0.07	0.03
TI	3.10	1.32	−0.15 **	−0.23 **	0.11 *	[0.84]	0.88	0.70	0.88	0.05	0.03

GEN = Gender, MAR = Marital status, EDU = Education, TEN = Tenure, POS = Perceived Organizational Support, OC = Organizational Commitment, JI = Job Insecurity, TI = Turnover Intention, α= Cronbach’s Alpha, CR = Composite Reliability, AVE = Average Variance Extracted, ASV = Average Shared Variance, MSV = Maximum Shared Variance, a = The square root of the AVE, Values in square brackets [] are the square root values of AVE, ** *p* < 0.01, * *p* < 0.05.

**Table 5 ijerph-19-08516-t005:** Results of testing the hypotheses.

**Model 1**	**Mediation Analysis**	**β**	**SE**	**t**	**LLCI**	**ULCI**	**Hypothesis**	**Result**
	**Outcome variable:** **TI**							
	Constant	3.65	0.38	9.60 ***	2.91	4.40	H_1_	Supported
	POS	−0.15	0.04	−3.42 ***	−0.23	−0.06		
	Gender	0.06	0.12	1.37	−0.07	0.41	R^2^ = 0.03 F_(5,484)_ = 3.40 *p* < 0.001	
	Marital Status	−0.08	0.13	−1.78	−0.48	0.02		
	Education	−0.05	0.06	−1.15	−0.19	0.05		
	Tenure	0.07	0.05	1.54	−0.02	0.18		
	**Outcome variable:** **OC**							
	Constant	1.40	0.30	4.59 ***	0.80	2.00	H_2_	Supported
	POS	0.66	0.03	19.03 ***	0.59	0.73		
	Gender	−0.01	0.10	−0.05	−0.20	0.19	R^2^ = 0.44 F_(5,484)_ = 76.45 *p* < 0.001	
	Marital Status	0.08	0.10	2.13 **	0.02	0.42		
	Education	−0.03	0.05	−0.86	−0.13	0.05		
	Tenure	0.01	0.04	0.35	−0.09	0.07		
	**Outcome variable:** **TI**							
	Constant	3.92	0.39	10.18 ***	3.16	4.68	H_3_-H_4_	Supported
	POS	−0.02	0.06	−0.42	−0.13	0.09		
	OC	−0.19	0.06	−3.25 ***	−0.30	−0.08		
	Gender	0.06	0.12	1.38	−0.07	0.40	R^2^ = 0.06 F_(6,483)_ = 4.78 *p* < 0.001	
	Marital Status	−0.07	0.13	−1.47	−0.44	0.06		
	Education	−0.06	0.06	−1.30	−0.19	0.04		
	Tenure	0.07	0.05	1.51	−0.02	0.17		
**Model 2**	**Moderation analysis**	**β**	**SE**	**t**	**LLCI**	**ULCI**	**Hypothesis**	**Result**
	**Outcome variable:** **OC**							
	Constant	4.43	0.27	16.27 ***	3.85	4.91	H_5_	Supported
	POS	0.57	0.04	15.83 ***	0.50	0.64		
	Infection Status (W)	−0.07	0.12	−0.56	−0.29	0.16		
	POS × W	−0.45	0.07	−6.79 ***	−0.58	−0.32		
	Gender	−0.01	0.09	−0.07	−0.19	0.17	R^2^ = 0.49 F_(7.482)_ = 66.27 *p* < 0.001	
	Marital Status	0.20	0.10	2.13 **	0.02	0.40		
	Education	−0.06	0.05	−1.22	−0.15	0.03		
	Tenure	0.02	0.40	−0.64	−0.10	0.05		
**Model 3**	**Moderation analysis**	**β**	**SE**	**t**	**LLCI**	**ULCI**	**Hypothesis**	**Result**
	**Outcome variable:** **TI**							
	Constant	2.88	0.34	8.50 ***	2.22	3.56	H_6_	Supported
	OC	−0.17	0.04	−4.17 ***	−0.29	−0.09		
	JI	0.09	0.04	2.24 *	0.01	0.16		
	(OC) × (JI)	0.11	0.02	4.91 ***	0.06	0.15		
	Gender	0.17	0.11	1.48	−0.05	0.41	R^2^ = 0.11 F_(7.482)_ = 8.22 *p* < 0.001	
	Marital Status	−0.18	0.12	−1.51	−0.43	0.06		
	Education	−0.04	0.06	0.78	−0.16	0.07		
	Tenure	−0.09	0.05	−1.81	−0.01	0.18		
**Conditional indirect effects of POS on TI ** **(POS** **→** **OC** **→** **TI)**			**β**	**SE**	**LLCI**	**ULCI**	**Hypothesis**	**Result**
	**Infection status**	**JI**						
**Model 4**	Infected	High	−0.37	0.07	−0.50	−0.24	H_7_	Supported
	Infected	Low	−0.02	0.06	−0.14	0.11		
	Not Infected	High	−0.19	0.04	−0.27	−0.12		
	Not Infected	Low	−0.01	0.03	−0.07	0.05	R^2^ = 0.11 F_(8.481)_ = 7.18 *p* < 0.001	
	Index of Moderated Mediation		−0.05	0.01	−0.08	−0.03		

*** *p* < 0.001, ** *p* < 0.01, * *p* < 0.05.

## Data Availability

The data presented in this study are available in anonymized form upon request from the corresponding author.

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
