# Peer review of "Living with Infection Risk and Job Insecurity during COVID-19: The Relationship of Organizational Support, Organizational Commitment, and Turnover Intention"

_ijerph, 2022, doi:10.3390/ijerph19148516_

Round 1
Reviewer 1 Report
The topic is exciting, and the proposed objective of the study has significant practical implications.
However, the consistency of the link between the supporting theoretical models (although well explained by the authors) and the social context analyzed is not clear to the readers.
Some of the hypotheses postulated in the paper are already well documented in the literature (H5: The effect of organizational support on organizational commitment differs by the infection status of employees; H3: Organizational commitment negatively affects turnover intentions; H2:Organizational support has a positive effect on organizational commitment). Reassessing them in this particular industry and crisis might be the real added value of this research. However, without an explanation of the social interaction influence on the situation and individual perception, one might find the hypothesis no H7 (The status of being infected and job insecurity play moderating roles on the indirect effect of perceived organizational support on turnover intentions via organizational commitment) as pure common sense… which is not.
Author Response
Dear Reviewer,
First of all, we would like to thank you for your support, and contributions to our paper and also positive comments. In line with your comments and suggestions, there are some new additions to the text as highlighted in red within the manuscript. Thank you for your valuable feedback.
Reviewer 2 Report
The manuscript is relevant and explores a little-known aspect such as the relationships within the context of perceived job insecurity and infection status of employees in the tourism sector, in addition to analyzing different variables of interest such as organizational support, organizational commitment and turnover intention.
It is a well-designed and developed manuscript; my main concern is its length; below, I indicate my comments about the work .
Lines 58-60 please restructure the sentence as it is difficult to understand.
Modify figure 1, some arrows look irregular. Likewise, it is recommended to briefly describe the figure.
The authors make an excellent review of the topic to substantiate different hypotheses; however, the introduction section is extensive, and it seems that it is more of a narrative review than an original manuscript in many scenarios. Therefore, the authors must assess whether it is essential to leave this section so extensive, which can distract the reader from the main objectives of the work.
Properly define the number of work centers involved in the study from the beginning of Material and Methods since it is not entirely clear.
In the material and methods section, the authors must indicate the type of sampling used in their study.
The authors must justify why an ethics committee or an IBR did not evaluate their study.
Authors must place a split line between columns 3 (%) and 4 (characteristics).
I recommend placing the figures immediately after their mention in the text.
The authors need to restructure the conclusion since, in addition to incorporating references (which is inappropriate), they do not adequately conclude with the most significant contributions of their work.
Author Response
Dear Reviewer,
First of all, we would like to thank you for your support, detailed assessments, and contributions to our paper, and also positive comments. Your recommendations helped us to assess more clearly of our study. In line with your comments and suggestions, there are some new additions to the text as highlighted in red within the manuscript. Also, our explanations/revisions are explained below.
Point 1: The manuscript is relevant and explores a little-known aspect such as the relationships within the context of perceived job insecurity and infection status of employees in the tourism sector, in addition to analyzing different variables of interest such as organizational support, organizational commitment, and turnover intention.
It is a well-designed and developed manuscript; my main concern is its length; below, I indicate my comments about the work.
Response 1: First of all, thank you for your valuable feedback and positive comments.
Point 2: Lines 58-60 please restructure the sentence as it is difficult to understand.
Response 2: In line with your suggestions, the expressions of the sentences were revised.
Point 3: Modify figure 1, some arrows look irregular. Likewise, it is recommended to briefly describe the figure.
Response 3: The arrows in Figure 1 have been revised. Also, explanations about Figure 1 have been made in the text.
Point 4: The authors make an excellent review of the topic to substantiate different hypotheses; however, the introduction section is extensive, and it seems that it is more of a narrative review than an original manuscript in many scenarios. Therefore, the authors must assess whether it is essential to leave this section so extensive, which can distract the reader from the main objectives of the work.
Response 4: Thank you for your valuable feedback. We prepared our study according to the instructions for the Authors of the International Journal of Environmental Research and Public Health. In the Instructions for Authors section, it was stated that the main text should be a minimum of 3000 words. The entire content of the text was prepared and revised in line with the issues stated by the journal.
Point 5: Properly define the number of work centers involved in the study from the beginning of Material and Methods since it is not entirely clear.
Response 5: In line with your suggestions, the number of work centers involved in the study was revised
Point 6: In the material and methods section, the authors must indicate the type of sampling used in their study. The authors must justify why an ethics committee or an IBR did not evaluate their study.
Response 6: In line with your suggestions, the sampling method and rationale were explained in the material and methods section of the study. At the same time, explanations about the ethics committee were included in the text.
Point 7: Authors must place a split line between columns 3 (%) and 4 (characteristics). I recommend placing the figures immediately after their mention in the text.
Response 7: In line with your suggestions, a split line was placed between columns 3 (%) and 4 (characteristics). Also, figures are given immediately after their mention in the text.
Point 8: The authors need to restructure the conclusion since, in addition to incorporating references (which is inappropriate), they do not adequately conclude with the most significant contributions of their work.
Response 8: In line with your suggestions, the conclusion section was revised and references were removed.
Thank you very much for your valuable contributions which helped us remarkably to improve the quality of our paper. And also thank you for your positive comments.
Reviewer 3 Report
Many thanks for the interesting work. At the same time, it seems to me that two issues need to be discussed.
1) Do you have data on the overall unemployment rate and the impact of the epidemic on it? Most of the employees are quite young, have a relatively low level of education, and, at the same time, about half of them are married. This may be a combination of factors that affect one’s anxiety about possible loss of a job. But in one way or another, it is necessary to understand the overall unemployment rate among people in this age cohort.
2) Please tell us about the wage system. Does the workload of hotels directly affect the earnings of workers?
Lines 515-530. 'These problems were experienced more severely by tourism workers, who are at increased risk of contracting COVID-19'. I would like to suggest adding information about healthcare workers who were considered to have a higher risk of psychological distress (10.3390/ijerph18020708 and 10.1016/j.genhosppsych.2020.06.007)
These clarification points can be discussed in the discussion section.
Overall, a good work was done.
Author Response
Dear Reviewer,
First of all, we would like to thank you for your support, and contributions to our paper and also positive comments. In line with your comments and suggestions, there are some new additions to the text as highlighted in red within the manuscript. Also, our explanations/revisions are explained below.
In line with your suggestions, in the discussion section, we have stated the unemployment rates in the tourism sector due to the pandemic. We explained it in the context of the demographic characteristics of the participants. Also, citing the relevant source, we added information about healthcare workers who were considered to have a higher risk of psychological distress
Thank you for your valuable feedback and positive comments.